# Is this a Real Choice? Critical Exploration of the Social License to Operate in the Oil Extraction Context of the Ecuadorian Amazon

**Alberto Diantini** [1,*]**, Salvatore Eugenio Pappalardo** [1]**, Tim Edwards Powers** [2]**, Daniele Codato** [1]**, Giuseppe Della Fera** [3]**, Marco Heredia-R** [4]**, Francesco Facchinelli** [5]**, Edoardo Crescini** [5] **and Massimo De Marchi** [1]

[1] Department of Civil, Environmental and Architectural Engineering, University of Padova, Via Marzolo, 35131 Padova, Italy; salvatore.pappalardo@unipd.it (S.E.P.); daniele.codato@unipd.it (D.C.); massimo.de-marchi@unipd.it (M.D.M.)
[2] Faculty of Education, Monash University, Melbourne 3800, Australia; tim.powers@monash.edu
[3] GIShub Association, 35138 Padova, Italy; giuseppe.dellafera@dicea.unipd.it
[4] Departamento de Ciencias de la Vida, Universidad Estatal Amazónica (UEA), Puyo 160101, Ecuador; mheredia@uea.edu.ec
[5] Interuniversity Department of Regional and Urban Studies and Planning (DIST), University of Turin, Viale Pier Andrea Mattioli, 10125 Torino, Italy; francesco.facchinelli@dicea.unipd.it (F.F.); edoardo.crescini@dicea.unipd.it (E.C.)
**\*** Correspondence: alberto.diantini@unipd.it

**Abstract:** The purpose of this research was to critically analyze the social license to operate (SLO) for an oil company operating in Block 10, an oil concession located in the Ecuadorian Amazon. The specific study area is an important biodiversity hotspot, inhabited by indigenous villages. A mixed-methods approach was used to support a deeper understanding of SLO, grounded in participants' direct experience. Semi-structured interviews (N = 53) were conducted with village leaders and members, indigenous associations, State institutions, and oil company staff, while household surveys were conducted with village residents (N = 346). The qualitative data informed a modified version of Moffat and Zhang's SLO model, which was tested through structural equation modelling (SEM) analyses. Compared to the reference model, our findings revealed a more crucial role of procedural fairness in building community trust, as well as acceptance and approval of the company. Procedural fairness was found to be central in mediating the relationship between trust and the effects of essential services provided by the company (medical assistance, education, house availability) and sources of livelihoods (i.e., fishing, hunting, harvesting, cultivating, and waterway quality). The main results suggested that the concept of SLO may not appropriately apply without taking into account a community's autonomy to decline company operation. To enhance procedural fairness and respect for the right of community self-determination, companies may need to consider the following: Establishing a meaningful and transparent dialogue with the local community; engaging the community in decision-making processes; enhancing fair distribution of project benefits; and properly addressing community concerns, even in the form of protests. The respect of the free prior informed consent procedure is also needed, through the collaboration of both the State and companies. The reduction of community dependence on companies (e.g., through the presence of developmental alternatives to oil extraction) is another important requirement to support an authentic SLO in the study area.

**Keywords:** social license to operate; trust; acceptance and approval; social performance; mixed methods; oil activities; procedural fairness; self-determination; corporate social responsibility; Ecuadorian Amazon

## 1. Introduction

*1.1. An Overview of the Concept of Social License to Operate and the Objectives of the Study*

The social license to operate (SLO) is usually defined as the level of acceptance an enterprise or a project has from the stakeholders, especially the local communities [1–4]. This implicit contract [3,5,6] reduces the social risk of a company (i.e., potentially costly conflicts with local communities) if its behavior builds trust with the communities: the higher the SLO, the lower the risk [7].

The concept of SLO has attracted much attention, especially for companies whose reputational risk is high [4,8–10]. This is relevant for extractive industries [1], where negative community effects can result in protest actions [11], damages to the company's reputation, project delays, and lost profits, reducing the access to future investment opportunities [12]. SLO has been associated with increased alignment with community expectations [8], heightened community engagement [13] on environmental and social issues [3], and more sustainable social developments within local communities [14]. It has strong links to corporate social responsibility (CSR), which refers to policies and practices (such as programs of social services) adopted by a company as a reflection of its commitment to local communities [15,16].

The SLO was initially raised in mining intensive developed countries, such as Canada and Australia, as a potential vehicle to influence corporate–community relations. It is now increasingly discussed in different countries (e.g., in the Latin America region) and across a wide range of contexts, such as forestry, mining, hydroelectric, and oil sectors (see for example [17–22]). However, Ehrnström-Fuentes and Kröger [23] observed SLO implementation challenges in the Latin American forestry context, including: the risk of co-optation, communities' dependence on companies for social services and job opportunities; the structural power imbalances between powerful stakeholders (e.g., the State and companies) and weak stakeholders (e.g., local populations); and the risk of hidden indigenous worldviews behind the dominant Western concept of development. According to our literature review, there has been little work assessing how SLO functions in the Latin American oil sector; one of the main drivers of the regional economy, but, at the same time, a cause of negative socio-environmental impacts and conflicts which threatens sustainable development [24–28].

To fill this gap in the literature, this research explored the function of SLO within community–company relations in remote indigenous villages of an oil production area of the Ecuadorian Amazon. Moreover, the study aimed to empirically model SLO in this context. Although SLO has been considered intangible [29–31] and difficult to measure [9,32], others have attempted to model the concept. For example, Moffat and Zhang's study [4], which focused on an Australian mining region, was the first to model the SLO critical elements. Their findings showed that establishing a trusting relationship between communities and mining companies was essential to achieve and maintain an SLO. Negative impacts on social infrastructures, communication quality between community members and mining company personnel, and procedural fairness in terms of giving communities a reasonable voice in the decision-making process, significantly affected the community's acceptance of mining activities through trust in the operating mining company [4]. Empirical measures may help companies gauge their attempt to gain community acceptance reflecting a social license to operate [33,34]. Moreover, an SLO measurement would quantify the social risk of the company's operations, providing a "consistent and robust benchmarking of social performance across time as an operation develops" (see [35] p. 484). As such, this approach is perfectly in line with the industry's language, which largely uses quantitative and empirical business terms in its narrative [5,15].

Moffat and Zhang's model [4] was taken as a reference to analyze how an SLO concept may apply to the remote Amazon. A mixed-method approach was utilized, combining semi-structured interviews and household surveys in order to provide a deeper understanding of the SLO process, suggest possible modifications to the model, and strengthen the study's conclusions. This approach mitigates possible limitations that can result from using a single quantitative approach [36,37]. This is the first mixed-methods study that critically investigated the concept of SLO in this context.

### 1.2. Elements of Social Licenses to Operate (SLO)

Drawing on findings from Moffat and Zhang's model [4] as presented in Figure 1, trust in the oil company is considered the most proximal predictor of acceptance and approval of the company's presence and operations. Trust, in turn, is influenced by procedural fairness, company–community contact, and impacts on social infrastructure. Compared to Moffat and Zhang's model, we proposed that trust may be influenced by a richer array of elements, which included indigenous perceptions of environmental quality and social aspects affected by company activities. The SLO elements are reviewed in the following sections. This literature review, in combination with the interview results, informed our modelling of SLO.

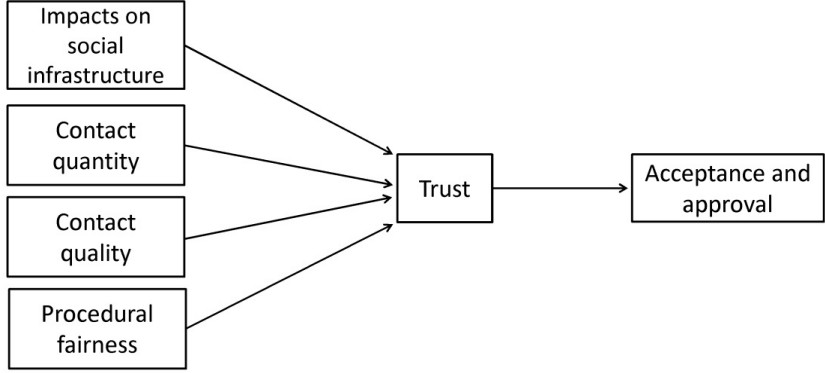

**Figure 1.** Moffat and Zhang's model [4].

#### 1.2.1. Acceptance and Approval

In the SLO literature, acceptance and approval are core outcomes of a social license, often framed as a continuum [7,38,39]. At one end, acceptance is the basic level, defined as the condition in which a community does not actively oppose a company or a project. At the other end of the continuum, approval is the condition in which communities actively show their support [8]. In this definition, acceptance is considered equivalent to "non-opposition." However, studies have argued some communities may not always openly express their dissent because it is considered culturally inappropriate [23,40]. Alternatively, conflict may simply be in a dormant phase, where communities are in the process of re-shaping their collective views, prior to open dissent [40–42]. Therefore, if approval presupposes active support, a superficial absence of conflict may not equate to acceptance [30,43,44]. As reported by others, the continuum may vary according to norms, social expectations, and behavior of the actors involved [7,38,39].

#### 1.2.2. Trust

Trust is central to SLO literature: full trust is related to high levels of acceptance and approval [1,2,4,8,10,45]. Trust in the company is generally presented as the condition in which the community expects that the company will keep its promises and seek mutual advantage [1,2] through various forms of collaboration and engagement [46,47]. Establishing trust is heavily dependent on community expectations coinciding with the expectations of the company [48,49]. In general, it can be difficult to build trust and even more difficult to rebuild once damaged [2]. For example, in the case of Hydro-Quebec in Canada, the Cree indigenous people's trust in the company was so poor in the 1970s, that it took over 30 years to establish a good relationship [2,50]. Companies that build and maintain trust tend to demonstrate honesty and integrity in their relationships with communities [18] through responsible management [4,34,51] by keeping community promises and agreements [31,34,52,53].

### 1.2.3. Procedural Fairness

Procedural fairness refers to the condition in which local communities perceive they have an active and respected role throughout the decision-making process, especially in relation to projects that may affect their territories [4,54–57]. Indeed, the engagement of local communities in the decision-making process is positively related to social acceptance [4,13,45,58].

Past studies reveal perceptions of procedural fairness are normally supported by transparency in providing local communities with open access to project information, including positive and negative aspects, such as economic benefits and socio-environmental impacts [4,30,59,60]. Such access to information empowers people in the decision-making process [61].

The theme of procedural fairness is closely related to the concept of free prior and informed consent (FPIC), which is often ignored by the SLO literature [5,18,62]. The use of FPIC was established under international law by the United Nations Declaration on the Rights of Indigenous Peoples (UNDRIP) and Indigenous and Tribal People Convention n.169. It has developed globally as an important mechanism to encourage fairer engagement with indigenous peoples concerning activities which may impact their territories [18,31,63,64]. In general, FPIC requires affected indigenous communities to be provided with adequate information about project risks and impacts prior to implementation. Further, they cannot be forced or coerced to give their consent [12,63,64]. Despite these international agreements, indigenous people in Latin America have continued to experience violations of FPIC procedures. Such violations have been often overlooked by governments, especially concerning projects of national interest, such as oil activities in Ecuador [65,66]. It has been argued that an SLO can only be developed when companies, together with governments, respect the principles of FPIC, not only at the beginning but throughout the project life-cycle [63,67,68].

A fair decision-making process also supports a distributive fairness, which refers to perceived equitable project benefits within or among communities [10,45,69]. In Amazonian contexts, the main benefits of extractive activities have been frequently directed only to a few beneficiaries (e.g., selected community leaders, families, or communities) instead of equal redistribution in the affected territory, causing discontent and conflicts [70,71]. Past research has demonstrated greater satisfaction when people believe they receive a fair share of benefits [10,45,72].

Procedural fairness appears to be central in supporting a social license. It is worth noting that procedural fairness was the strongest predictor of trust in Moffat and Zhang's original model [4]. Additionally, procedural fairness was strongly related to other predictors of trust (i.e., impacts on social infrastructure, contact quality, and quantity), suggesting procedural fairness may hold a more prominent role in building trust.

### 1.2.4. Contact

Continuous and transparent dialogue between companies and communities is a fundamental element supporting an SLO [73,74]. Positive forms of contact and interactions have improved inter-group relations [75,76], especially in the extractive context, where communities are generally skeptical of the benefits. In such instances, community–company contact appears critical to maintaining good relations [8]. Based on previous studies [75,77], Moffat and Zhang [4] distinguished quality and quantity of contact. In their study, only contact quality was a significant predictor of community trust, whereas contact quantity was not. Contact quality was more strongly related to procedural fairness than trust (see Figure 2 in [4]), which suggested that higher contact quality was related to higher procedural fairness.

### 1.2.5. Environmental Quality

Considering the importance of the environment to indigenous populations [78,79], the development of an SLO seems highly dependent upon a company's ability to avoid or mitigate environmental impacts. Companies that demonstrate respect for environmental protection standards appear to positively

influence the community's trust in the company itself [4,32,80], especially for indigenous communities [81]. Companies in extractive operations that have demonstrated competence and a willingness to reduce environmental impacts, have reported higher levels of community acceptance and approval [4,45,82]. This topic may inform the conceptualization of SLO.

A wide range of negative environmental impacts as a consequence of oil activities has been reported in the western Amazon [27,83–87], where these activities are concentrated [27,88,89]. As suggested by Diantini [90], the environmental impacts of oil extraction can be summarized under the following categories: (1) Atmosphere, (2) acoustic environment, (3) aquatic environment, (4) soil, and (5) flora, fauna, and ecosystems.

Most atmospheric impacts are related to gas flaring, that consists of on-site waste gas burning [91]. This activity releases toxic combustion gases and dust that may affect the population's health [92,93]. It is a common practice in the Ecuadorian Amazon where an estimated 6888 MCM of gas was burnt from 102 sites during the period of 2012–2018 [91]. Other atmospheric impacts are related to gas combustion emissions from power generators and the first processing phases of crude oil [90]. Oil extraction operations are also the source of acoustic emissions and vibrations. Examples include explosive blasting in the exploration phase, well drilling, gas flaring practice, movement of heavy equipment, and the extensive use of power generators [90].

Aquatic environmental impacts are related to surface and underground water contamination, due to oil spills from pipelines and waste release (e.g., water formation, drilling muds, and other toxic liquids) into rivers or ponds without any protection measures [27,94,95]. In the Ecuadorian Amazon, the use of obsolete technology and low standards of environmental protection measures have been linked to disastrous spills, with high impacts on the quality of water sources [85,96].

Atmospheric and aquatic environmental impacts can also generate negative effects in the soil and subsoil matrix. For example, gas flaring dust, oil spills, and intentional or accidental discharges of toxic fluids can cause the alteration of the chemical and microbiological characteristics of the soil, potentially lasting decades in the absence of effective environmental restoration [97,98].

The impacts on the atmosphere, water, and soil are directly related to multiple effects on flora, fauna, and ecosystems [84,99]. For example, oil contamination has led to bioaccumulation and biomagnification of hydrocarbons and metals in the food chain [100,101]. In the Amazon, frequent toxic spills are reportedly threatening biodiversity and ecosystems [27,102,103].

Given the substantial impact of oil operations on the environment, we surmise that an SLO is linked to community environmental perceptions. This is likely to be particularly sensitive to indigenous communities with historically strong connections to the environment [78,79].

### 1.2.6. Social Aspects

In Moffat and Zhang's model [4], community perceptions regarding the impacts on social infrastructures (such as local hospital capacity and housing availability and affordability) were found to have an important role predicting trust. We argue that social perceptions can extend beyond infrastructure to encompass community perceptions of social impacts and the quality of social services offered by the company to communities.

A wide range of negative social impacts resulting from company activities can be decisive in undermining community trust and acceptance [4,12,72]. Examples from the Amazon include, the reduction of communities' access to natural resources, such as: water to drink, cook, or wash; soil to cultivate; and forest animals and plants for sustenance [83,104]. These represent critical threats to indigenous people's livelihood and practices, such as the collection of wild vegetation, the practice of small-scale agriculture, hunting, and fishing [83,100,105,106]. Other social impacts regard health implications associated with oil pollution, ranging from respiratory diseases due to the combustion compounds in gas flaring activities, to dermatitis, diarrhea, gastritis, birth defects, miscarriages, and cancer [85,107,108].

Social services are often provided by companies to their host communities, as a form of compensation. These social services generally include support in the fields of health care, education, food provision,

job opportunities, agriculture, sport, and transport infrastructure [71,109,110]. Such practices are quite common in the Amazon and other similar contexts with low levels of socio-economic development, as part of companies' corporate social responsibility program. If these programs are properly managed they can lead to long-term and sustainable community development, helping the company gain greater acceptance [30,111]. In some cases, the company's provision of social services may overlap with State responsibilities with a deleterious effect of increasing the community's dependence on the company, without socio-economic improvement [68,109].

We argue companies that adequately manage social impacts and promote social services, are more likely to engender trust, and approval of a company's presence. We argue this expanded view of social aspects (beyond infrastructure) will inform our understanding of SLO.

## 2. Materials and Methods

This paper draws on fieldwork and data collected during 2018–2019 in remote villages of Block 10, of an oil concession located in the Central-Southern Ecuadorian Amazon (see Figure 2), with high biodiversity which is critically valued by conservationists [86,112]. The villages are inhabited by an indigenous population, mostly Kichwa. The oil company that operated in the block was Agip Oil Ecuador (since renamed Agip), which provided social services for local villages as a compensatory measure for its activities.

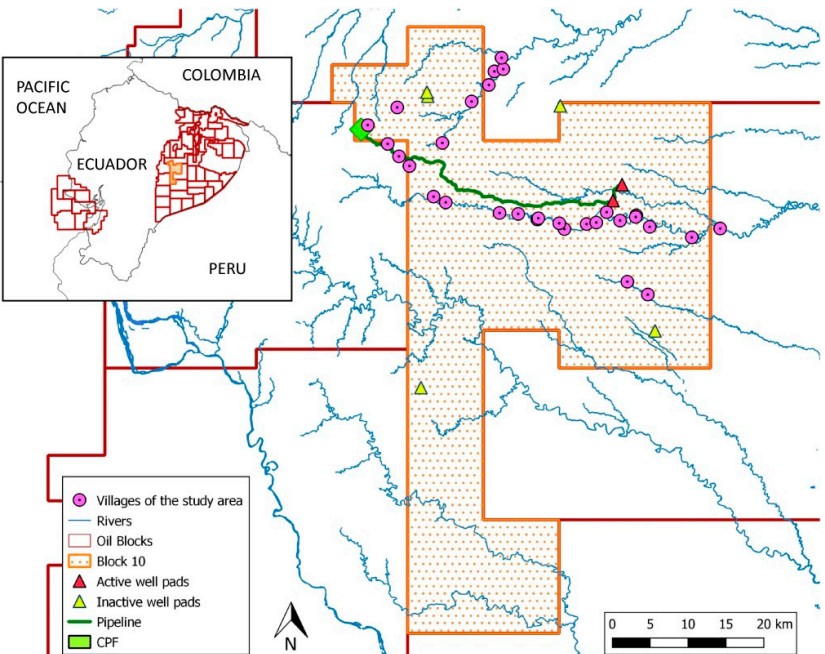

**Figure 2.** Location of participating villages within Block 10, in the Central-Southern Ecuadorian Amazon.

The methodology of the study was based on a mixed-methods approach, involving qualitative (interviews) and quantitative (questionnaire) analyses. The qualitative data was used to inform a modified version of Moffat and Zhang's model [4]. The mixed-methods approach was also used to increase confidence in findings by utilizing different methodologies to cross-check data [113] for greater confidence in robust results [114]. The semi-structured interviews and household surveys were conducted, independently of the company, by the research group, composed of the first author and 10 indigenous students from the local university (Universidad Estatal Amazónica de Puyo). The following describes the interview process, followed by the questionnaire design, measures, survey procedure, and analytical approach.

*2.1. The Interview Process*

In the period January–March 2018, the study began with preliminary semi-structured interviews with village leaders and residents, indigenous associations, non-governmental organizations (NGOs), local researchers, representatives of the Ministry of Environment and Ministry of Hydrocarbons, and company personnel. The village leaders and residents were selected through using a snowball sampling technique [115]. The interview questions specifically asked respondents about their views on company-provided services, past incidents (e.g., oil spills) and their management, socio-environmental impacts, reasons that instigated community protests, and views on new proposed oil projects. In addition, these interviews informed the development of the household survey conducted during March–April 2018. Additional semi-structured interviews with village leaders and members, indigenous associations, NGOs, and the company were conducted after the household survey to explore more deeply aspects that emerged during the survey, such as cases of spills mentioned by participants and the company's response to protests. The interviews ($N = 53$) before and after the household survey were analyzed in combination.

*2.2. The Questionnaire Design and Measures*

2.2.1. The Questionnaire Design

The questionnaire targeted village residents (almost all of them were indigenous) within the study area to collect their views on the proposed SLO elements. Both closed- and open-ended questions were included in the questionnaire giving respondents the opportunity to provide added details. The questionnaire structure can be defined as a "mixed concurrent design" (or concurrent triangulation design), where quantitative and qualitative data contributed simultaneously to produce same-sample information which can be cross-checked [116].

The questionnaire was developed on the basis of the literature review and information collected during the preliminary semi-structured interviews. The household questionnaire was piloted in a single village before wider distribution. The piloted questionnaire resulted in modifications or removal of items which were not relevant to the local circumstances (e.g., the access to potable water as a service provided by the company, the potential impacts on livestock breeding, and the impact on tourism). Contact quality and contact quantity were eliminated because it became clear that few village residents had contact with the company (normally only the village leaders had contact). Other changes focused on culturally appropriate language to improve comprehension. In addition, the questionnaire was translated from Spanish to the local indigenous language, Kichwa, by the indigenous university students involved in the data collection.

2.2.2. The Measures

The final version of the questionnaire was composed of 35 items which measured five latent constructs. Also included were 11 opened-ended questions. See Appendix A for a complete set of questions.

Environmental quality was measured with seven items developed for this study. Participants, considering the presence of the oil company in the territory, were asked about the perceived environmental changes, including: air, water bodies (rivers, slopes, and lagoons), soil, flora, fauna, noise intensity, and night light intensity (e.g., *How do you think these environmental compounds are changing in the proximity of your village in relation to the presence of the oil company?*). The response scale ranged from *1 (much worse)* to *5 (much better)*.

Social aspects were measured with 14 context-specific items developed for this study. Particularly, seven items investigated perceived changes attributed to the presence of the company in: hunting, fishing, gathering forest products, cultivating, the quality of water for drinking, cooking, and washing. The remaining seven items examined perceptions of the changes in: access to electricity, recreational services (soccer or volleyball fields, etc.), medical assistance, education, transportation infrastructure, housing, and employment opportunities (e.g., *How do you think these aspects are changing for your village*

*in relation to the presence of the oil company? Medical assistance*). The response scale was *1 (much worse)* to *5 (much better)*.

Procedural fairness was measured with eight items. Two were adapted from Moffat and Zhang [4], originally from Tyler [57], reflecting the company's responsiveness to village opinions (e.g., *To what extent do you agree with the following statements? The oil company listens to and respects the village's opinions*). The other six items were developed for this study to capture equitable distribution of benefits and FPIC-related concepts, such as prior information on projects' impacts and community prior consent (e.g., *To what extent do you agree with the following statements? Before the implementation of new oil operations, the village is always informed about risks and impacts*). The response scale was *1 (strongly disagree)* to *5 (strongly agree)*.

Trust was measured with four items, largely based on Moffat and Zhang's measures [4], adapted from Tam et al. [117]. Items were adjusted to reflect cultural norms and improve clarity (e.g., *Do you trust that the oil company is doing everything possible to avoid impacts on the environment?*). A new item was developed to investigate the level of trust that the company keeps its promises (e.g., *Do you trust the oil company to keep its promises?*). The response scale was *1 = total distrust* to *5 = absolute trust*.

Acceptance and approval was measured with two items from Moffat and Zhang's [4] (e.g., *To what extent do you accept the presence of the oil company in your territory?*). The scale ranged from *1 (not at all)* to *5 (very much)*.

The open-ended questions were utilized to qualitatively explore some aspects measured by items. An open-ended question was utilized for each item measuring environmental quality. One was developed for social aspects, concerning the quality of plantations. For procedural fairness, three open-ended questions were prepared to investigate: (1) participants' knowledge of prior, free, and informed consent; (2) community coercion or intimidation; and (3) company-provided information about the state of the territory.

## 2.3. The Questionnaire Procedure

The household questionnaire was conducted in the villages of the study area. The survey sample frame included the total population in the study area estimated to be 3800–4000 residents. Our targeted survey sample size was 20% of the adult population (over 18 years of age), which was estimated to be 350–400 people. Most individual questionnaires were conducted with one or two residents for each household visited. Some participants were surveyed right after introductory meetings which explained the research to the community, for convenience (i.e., they lived far away) or due to work commitments. The questionnaire was read by the researchers to the participants in order to facilitate comprehension and overcome prominent literacy issues. Most surveys were conducted in Spanish and, sometimes, in Kichwa, to ease the expression of daily aspects. The verbal approach enhanced the qualitative information collected in the open questions.

It is noteworthy that in the initial phase of the survey, participants expressed their concern that the company would have cut social programs altogether if they took part in the study. In this situation, a signed informed consent sheet was not used, but we explicitly applied the principles of ethical research [118], discussing objectives, risks, methodology, and timeline of the study with village members. One of the pillars of our positive relationship with participants was the presentation of questionnaire results. The restitution of results is an ethical duty often overlooked in conducting human research, especially when indigenous communities are involved [119,120].

## 2.4. Analytical Plan

Firstly, we analyzed the qualitative data collected through the open-ended questions of the questionnaire and the semi-structured interviews. The qualitative information was transcribed and analyzed through Atlas.ti 8 [121] by the first author, using a grounded theory methodology [122]. Open coding was initially employed to segment the data, in order to label concepts, and later define and develop categories, their respective links, and the relation with SLO key themes. The results

of the qualitative analysis and the literature review informed the development of our hypothetical SLO model as a modification of Moffat and Zhang's [4]. Structural equation modelling (SEM) was employed to analyze the model, using AMOS 24 [123]. SEM is a useful framework for modelling latent constructs, explicitly allowing for measurement error, and testing overall model fit [124]. Confirmatory factor analyses (CFA) were conducted on the measurement and structural models. Goodness-of-fit was assessed using: the $\chi^2/df$ ratio, the comparative fit index (CFI), the Tucker–Lewis index (TLI), the root mean square error of approximation (RMSEA), and the standardized root mean square residual (SRMR). The $\chi^2$ index was included for completeness, although recognized as sensitive to sample size [124]. CFI and TLI values > 0.95, RMSEA < 0.60 and SRMSR < 0.08 describe good model fit [125]. Misfit in models were explored via modification indices (MI) and exploratory factor analysis (EFA), in order to identify potentially useful alternative factor measurement structures [126,127]. Appropriate use of EFA was tested by the Kaiser–Mayer–Olikin (KMO) as well as Bartlett's test of sphericity [128–130]. Screen plot and eigenvalues were used to determine the number of retained factors [131]. Factor items were eliminated if their loading was not important (less than +/−0.30) [131]. Reliability of factors was assessed using Cronbach's alpha [132] whereby factors with low reliability ($\alpha < 0.5$) were discarded, while those with moderate reliability (greater than 0.5) were retained [133].

## 3. Results

Questionnaires were collected from 375 participants. The sample was reduced by 29 participants ($N = 346$) due to item missingness (listwise deletion, see [126,127]). The average age of participants was 38 (SD = 14.11), and predominantly male (59%). Regarding education, 14.5% participants were illiterate, while 42.5% had a primary education, 36.1% had secondary education, 4.9% had post-secondary education (professional education, bachelor's degree, or more), while 2% declined to offer their education level. Contact quality and quantity were excluded from the analysis because, as previously mentioned, interviews and the piloted survey indicated few village members (primarily only village leaders) had contact with the company.

### 3.1. The Qualitative Information

In this section we present the qualitative results from the questionnaire open questions and the semi-structured interviews. Comments revealed a strong link with the procedural fairness themes including: community information about the state of the environment, FPIC topics (prior information and consultation, village's participation in the decision-making procedure of company's projects), distribution of company benefits among villages, and the reduction of essential services. Time and again, interviewees emphasized the importance of fair treatment in regard to their interactions with the company.

One of the most problematic issues that emerged was the difficult access to environmental information. Village residents and leaders frequently stated that the company did not inform the villages regularly about the quality of the environment, despite company declarations all information was available at the Ministry of Environment. Participants affirmed a lack of trust in the company because it had not provided easy access to information. This was reflected in perceived tension between residents, the company, and the Ministry, in regard to access to information. For example, upon our enquiries about a past spill, the company directed us to find the information at the Ministry. Upon contacting the local offices of the Ministry, we were informed the documentation was located at the central offices, where, instead, we were told no document was available. Finally, no enquiry produced results. Residents felt greater transparency in the management of environmental information was needed from both the Ministry and the company. Residents felt the company closely monitored and controlled what information was released to the community. Perhaps indicative of this control, the company closely monitored our activities in the field. Surprisingly, in the first meeting we had with the company, company representatives had a draft version of the questionnaire we were still developing and had not yet released.

Many respondents complained about a lack of consultation prior to recent company developments, especially those in closest proximity to the completed project. In this regard, the company stated prior consultation was a responsibility of the State, despite residents' interest in direct consultation with the company. Concerns over consultation extended to other activities managed by the company, including the provision of services. In particular, residents resented instances where the company threatened reduction or redirection of funding for services to other communities if residents did not support company decisions, or displayed open dissent (e.g., protests). These issues were considered quite unfair among the villagers, regardless of their proximity to the company facilities which led us to believe procedural fairness was a particularly important and prominent theme raised by participants.

Regarding essential services provided by the company (such as medical assistance, education, housing availability), the most commonly cited issue raised by participants was the reduction of these services over time. Interviews with officials of the Ministry of Hydrocarbons and the company suggested this reduction was associated with national contract changes between the State and the oil companies in 2010. The modified contract shifted responsibility for community social services to the State, leading companies to gradually reduce their involvement. However, interviewed residents felt the State had not met this responsibility. Many residents held expectations that extraction companies should provide these essential services, as they once used to, as a fair condition to maintain operations. Many participants complained about the lack of village engagement in the decision-making process which led to this shift in responsibility.

There were strongly held views that the environment was widely affected by company operations. There was a perceived reduction in air quality, especially in locations close to the Central Process Facility (CPF, the plant of first treatment of crude oil), and the well pads. Participants expressed concern that air pollution would negatively affect the quality of their crops and personal health as a result of combustion dust. Noise and night light pollution was considered problematic mostly in the proximity of the CPF and the extractive pads. Here residents believed the noise and the night light caused animals to migrate away from this area. Many participants perceived that flora and fauna were possibly affected by the oil operations, due to the contamination of the environment.

Other comments revealed themes regarding the perceived contamination of rivers and streams, reported cases of spills and resultant impacts on farming and fishing. Interviewed residents reported high levels of polluted waterways as a result of combustion dust and high impact events, such as oil spills and uncontrolled toxic chemical fluid discharge. Many declared that bathing in the rivers was no longer recommended due to contamination, which had been blamed for elevated cases of dermatitis and other skin diseases, especially in children and elderly people. With respect to residents' livelihood activities (e.g., fishing, hunting, cultivating), in participants' opinions, the most impacted was their typical small-scale traditional farming techniques. Interviewed residents perceived the quality of products like *yuca* (manioc), papaya, and banana had deteriorated as a consequence of environmental contamination since the company started oil extraction (late '90s). Residents alleged this problem would continue to affect peoples' health and the economic viability of traditional farming. Another activity considered as compromised was fishing, due to the perceived pollution of rivers. Hunting and harvesting forest products were generally considered less affected, since these activities were often carried out further away from oil facilities.

Water used for drinking, cooking, and washing depends on location: In certain areas, residents carry water from less polluted tributary streams far from the oil facilities; however, in areas closer to the CPF, residents complained they could not use stream water due to the pollution and relied on rainwater, although this too was perceived as contaminated by combustion dust.

Considering trust, the most common themes related to profit-driven development and lack of socioenvironmental protection measures. Many participants felt they had been "swindled" by the company, given the initial promised services and job opportunities were not always realized. Even in the villages with relatively higher benefits, participants frequently complained that the company had not fulfilled promises. Interviewees reported pervasive distrust towards the company attributed to

insufficient measures to protect the environment. In addition, a number of interviewees felt that the company had attempted to generate conflicts within and between villages, in an attempt to control coordinated protests. However, some residents expressed higher levels of trust, but these individuals were more commonly located in villages with higher levels of company-provided services and further away from oil extraction facilities.

Finally, acceptance and approval was generally quite low amongst interviewed residents. A general feeling of dissatisfaction was expressed by many participants towards their lack of involvement in the decision-making process, promised but unrealized development, decreased company-supplied social services, negative environmental and social impacts as a result of company activities, and the restraint imposed on open dissent. Notwithstanding this low level of acceptance and approval, only a few participants expressed a desire for the company to leave the territory because of the impacts it caused.

*3.2. The Quantitative Analysis*

Based on the analysis of the qualitative data, procedural fairness was elevated to a more important role than originally suggested by Moffat and Zhang's model. Interviewee comments often related to procedural fairness, such as active community engagement in the decision-making process and access to information. Since procedural fairness topics were mentioned also in relation to elements such as company's social services and environmental quality perceptions, we suspected that procedural fairness had a primary role in mediating the effects of perceptions related to social aspects and environmental quality on trust. On this basis, we proposed the hypothetical model displayed in Figure 3, tested through SEM analysis.

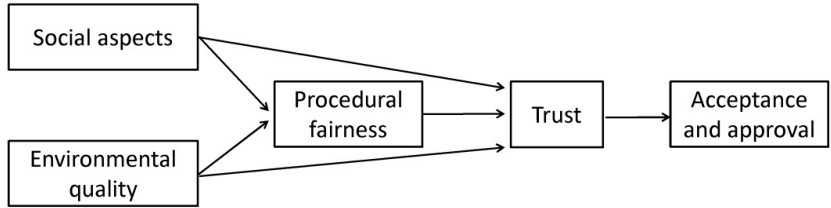

**Figure 3.** The social license to operate (SLO) hypothetical model.

A CFA was conducted to affirm the measurement of the five factors: Environmental quality, social aspects, procedural fairness, trust, and acceptance and approval. The measurement model (with factors covaried) did not fit the data well ($\chi^2$/df ratio = 2.490, $p$ = 0.00, CFI = 0.801, TLI = 0.785, RMSEA = 0.064, SRMR = 0.097). This indicated an EFA may be informative.

The EFA results (Table 1) suggested eliminating one item due to low loadings across all potential factors. The resultant EFA suggested a sound representation of factors (KMO = 0.818; Bartlett's test = 2249.40, $p$ < 0.001) in a simple structure with six factors, which explained 61.05% of the variance. The six factors measured:

- Water use: Quality of water to drink, cook, and wash;
- Sustenance sources: Livelihood activities (fishing, hunting, cultivating, and harvesting) and quality of rivers and streams;
- Flora and fauna quality: Quality of soil, flora, and fauna;
- Atmospheric quality: Night light and noise intensity and quality of air;
- Essential services: Services provided by the company, such as medical assistance, education, house availability;
- Support services: Services, more frequently managed by the State, such as recreational services, transport infrastructure, electricity access.

Support services was eliminated from the model because of low reliability ($\alpha$ = 0.45) and low prevalence; such services were rarely provided by the company in these communities. However,

essential services and atmospheric quality, despite their moderate reliability, were retained in accordance with qualitative information that indicated these factors were important to the study's aims.

**Table 1.** Mean, standard deviation, factor loadings, and Cronbach's alphas for 'Social Aspect' and 'Environment Quality' items.

| Items | M (SD) | Water Use | Sustenance Sources | Flora and Fauna Quality | Atmospheric Quality | Essential Services | Support Services |
|---|---|---|---|---|---|---|---|
| Water to drink | 2.92 (0.64) | 0.917 | | | | | |
| Water to cook | 2.95 (0.63) | 0.973 | | | | | |
| Water to wash | 2.96 (0.62) | 0.908 | | | | | |
| Fishing | 2.58 (0.66) | | 0.845 | | | | |
| Hunting | 2.80 (0.57) | | 0.559 | | | | |
| Cultivating | 2.29 (0.62) | | 0.469 | | | | |
| Harvesting | 2.89 (0.61) | | 0.527 | | | | |
| Rivers and streams | 2.36 (0.60) | | 0.522 | | | | |
| Soil | 2.25 (0.53) | | | 0.362 | | | |
| Flora | 2.67 (0.52) | | | 0.966 | | | |
| Fauna | 2.52 (0.53) | | | 0.530 | | | |
| Air | 2.42 (0.60) | | | | 0.406 | | |
| Night light | 2.83 (0.41) | | | | 0.640 | | |
| Noises | 2.53 (0.66) | | | | 0.594 | | |
| Medical assistance | 2.55 (0.87) | | | | | 0.786 | |
| House availability | 3.04 (0.67) | | | | | 0.359 | |
| Education | 3.17 (0.88) | | | | | 0.353 | |
| Recreational services | 2.98 (0.37) | | | | | | 0.528 |
| Transport infrastructures | 3.13 (0.57) | | | | | | 0.462 |
| Electricity access | 3.02 (0.33) | | | | | | 0.501 |
| Explained variance Tot. 61.05% | | 24.00% | 10.18% | 8.80% | 6.86% | 5.87% | 5.34% |
| Cronbach's alphas | | 0.82 | 0.71 | 0.70 | 0.60 | 0.52 | 0.45 |

A separate CFA was conducted on the measurement model with the updated set of factors, including water use, sustenance sources, flora and fauna quality, atmospheric quality, essential services, procedural fairness, trust, and acceptance and approval. Modification indices suggested covarying items with high theoretical connection: two items within procedural fairness (prior community consent about projects was covaried with changes in the projects based on community's opinions before their implementation); and two items measuring trust (company's behavior about avoiding impacts on the environment was covaried with company's behavior about avoiding social impacts on the village). With these covariances added, the model represented the data well ($\chi^2$/df ratio = 1.411, $p$ = 0.00, CFI = 0.961, TLI = 0.955, RMSEA = 0.035, SRMR = 0.047).

The proposed model was updated with the confirmed measures, as represented in Figure 4. The model presented strong fit measures ($\chi^2$/df ratio = 1.438 $p$ = 0.00, CFI = 0.956, TLI = 0.950, RMSEA = 0.036, SRMR = 0.049).

The means of all factors were below 3.00 on a scale of 1 to 5 (see Table 2); the highest was water use (M = 2.94, SD = 0.60) and the lowest was atmospheric quality (M = 2.48, SD = 0.42). This signified generally low-level perceptions across all measures including the outcome measures of trust (M = 2.39, SD = 0.63) and acceptance and approval (M = 1.79, SD = 0.76).

Acceptance and approval was most highly correlated with trust (*r* = 0.62), followed by procedural fairness (*r* = 0.40) while moderately related to all other factors (*r* ranging from 0.16 to 0.29). Trust was strongly related to procedural fairness (*r* = 0.65) and essential services (*r* = 0.47).

Trust in the oil company was a strong predictor of acceptance and approval of the company's presence (β = 0.62) and procedural fairness was a strong predictor of trust (β = 0.50) (Figure 4). The relationship between sustenance sources and trust was mediated by procedural fairness. As such, higher ratings of sustenance sources raised perceptions of procedural fairness which, in turn, predicted higher levels of trust. Essential services was partially mediated by procedural fairness, since it predicted procedural fairness (β = 0.31), as well as predicted trust (β = 0.25). Although the bivariate relationship between essential services and trust was high (*r* = 0.47), a substantial part of this positive relationship

was explained via heightened perceptions of procedural fairness. Perceptions of atmospheric quality, flora and fauna quality and water uses did not predict either procedural fairness or trust in the company.

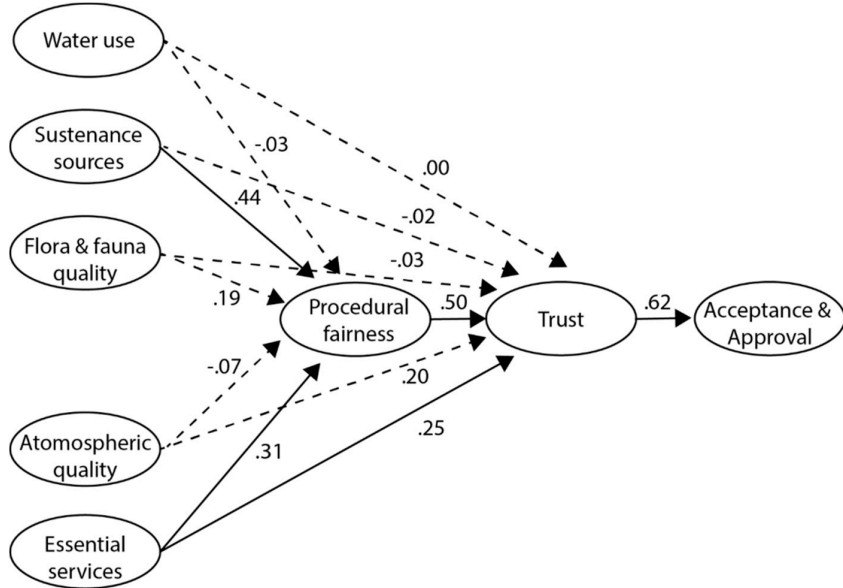

**Figure 4.** The revised social license to operate model. *Note.* Regression parameters are standardized. Solid lines represent significant relationships ($p < 0.05$); dashed lines represent non-significant relationships.

**Table 2.** Descriptive statistics and correlation matrix.

| Factors | M (SD) | 1 | 2 | 3 | 4 | 5 | 6 | 7 |
|---|---|---|---|---|---|---|---|---|
| 1. Procedural Fairness | 2.77 (0.64) | | | | | | | |
| 2. Water Use | 2.94 (0.60) | 0.28 ** | | | | | | |
| 3. Atmospheric Quality | 2.48 (0.42) | 0.37 ** | 0.32 ** | | | | | |
| 4. Flora and Fauna Quality | 2.60 (0.46) | 0.52 ** | 0.33 ** | 0.68 ** | | | | |
| 5. Sustenance sources | 2.58 (0.42) | 0.57 ** | 0.46 ** | 0.65 ** | 0.69 ** | | | |
| 6. Essential Services | 2.92 (0.58) | 0.43 ** | 0.27 ** | 0.13 | 0.30 ** | 0.20 * | | |
| 7. Trust | 2.39 (0.63) | 0.65 ** | 0.25 ** | 0.38 ** | 0.42 ** | 0.41 ** | 0.47 ** | |
| 8. Acceptance and Approval | 1.79 (0.76) | 0.40 ** | 0.16 ** | 0.23 ** | 0.26 ** | 0.26 ** | 0.29 ** | 0.62 ** |

$p < 0.05$ *; $p < 0.01$ **.

## 4. Discussion

### 4.1. An Overview of the Social License to Operate Emerging from the Study

The social license to operate (SLO) depicted by our model showed that trust, as a precondition of acceptance and approval, was essentially explained by procedural fairness, essential services, and sustenance sources. Compared to other studies [4,10,45], our findings suggested that procedural fairness had a more important role in the SLO process. It is not simply a predictor of trust but mediates the effects of sustenance sources and essential services (partially) on trust.

Regarding the procedural fairness themes, the qualitative data highlighted the importance local communities place on their opportunity to participate in decision-making processes for both new oil activities and the management of social services. Where meaningful dialogue did not occur, participants expressed dissent through forms of protest, to voice their opinions, concerns, and disagreement [11,67]. Protests have been interpreted as a part of a dialogic process seeking participation in decision-making [41,42,67,134]. Participants in this study reported that the company often controlled protests through retaliatory procedures and generating conflict within and between villages. The retaliatory procedures and the *divide et impera* ("divide and conquer") strategy are well-documented techniques,

often used by companies working in different sectors (mining, oil extraction, hydroelectric, etc.) to reduce and hide conflicts [67,71].

Consistent with the SLO literature [4,30,59,60], transparency was highly valued by participants as a key aspect of procedural fairness. Transparency appeared to be supported when the company informed communities about the state of the environment, along with the possible risks and benefits concerning the company's projects, prior to and during their implementation. When residents perceived a lack of clear transparent engagement regarding projects, protests appeared more likely as a means to have their concerns addressed [2,11,67].

This study demonstrated the provision of essential services such as health, educational programs, and housing directly affected both perceptions of procedural fairness and trust in the company, which sequentially predicted acceptance and approval of company activities. Given regulatory changes in 2010 which shifted responsibility for providing such services from the company to the State, there appeared to be growing disappointment amongst residents aimed at the company regarding the reduction of essential services. This situation caused conflicts between villages, the company, and the State, where residents felt unfairly treated, disengaged with the decision-making process, which negatively influenced trust in and acceptance and approval of the company. Residents' expectations were also undermined by limited job opportunities. Residents reported feeling deceived by the company when it did not keep promises of providing social services and job opportunities. This general sense of dissatisfaction, disillusion, and resignation in the face of promised services and job opportunities appears quite common when oil operations take place in areas that were poorly developed from a socioeconomic point of view [135–137]. These negative feelings were less common in some study villages which were under current negotiation for future oil infrastructures: residents hoped new oil activities would lead to more benefits.

Although the environment was perceived as widely contaminated by the company's activities, on our final SLO model, water use and perceived atmospheric quality and fauna and fauna quality did not directly affect either procedural fairness or community trust. This may be explained through the qualitative information, which highlighted environmental elements (in particular soil, air, and water bodies) were most concerning for residents because of their impact on their livelihood and health. In line with these findings, sustenance sources (cultivating, fishing, hunting, harvesting, and quality of waterways) were found to have an important role in predicting trust through the mediation of procedural fairness. This means that the perceived negative effects on sustenance sources lowered participants' perception of procedural fairness. This was consistent with participants' request for transparent management of environmental information and adequate environmental controls, in order to establish fair community engagement concerning environmental issues.

Expressed perceptions of extensive environmental contamination within this study are similar to those reported by residents in other oil areas of Centre-North of the Ecuadorian Amazon, where a 50-year history of oil extraction has been implicated in major environmental impacts and negative health effects [85,96,138,139]. Despite these historical learnings, there has been little improvement in environmental controls, recognition, or compensation resulting from the detrimental effects of oil activities [138,139].

Regardless of reported low levels of trust and acceptance and approval, residents expressed continued interest in social programs provided by the company. This dependency of the villages on the company, was quite similar to what Arsel et al. [140] defined as "Maria's Paradox", describing an Ecuadorian resident's divergent view which supported an oil company's provision of community services while also opposing the environmental impact. As in other disadvantaged remote areas of the Ecuadorian Amazon [109,110] and in our study area, low levels of socioeconomic status have permitted oil companies to operate despite low levels of acceptance. Indeed, the absence of the State in such regions has caused a short circuit of responsibilities: Residents feel the State has not addressed their essential service needs and relinquished such responsibilities to oil companies as a fair exchange for oil extraction. This condition has legitimized oil operations despite residents' perception

of detrimental environmental and social impacts [109,110,140]. Moreover, the chronicled dependence of remote communities on oil companies has been exacerbated by the developmental policies of Ecuador which, similar to other Latin American countries, has intensified the exploitation of natural resources, such as hydrocarbons and minerals [28,140]. These extractivist policies have depicted oil extraction industry as indispensable to the national economy and local development, impeding the enactment of economic alternatives [140,141]. Given the remote location, the villages in the study area appeared to identify the company as the only partner available to negotiate for services. However, the company's strategy to suppress local dissent seemed to have resulted in relatively low levels of conflict, which may inaccurately be interpreted as an SLO. Consistent with the literature [30,43,44], this study indicated an apparent absence or reduced presence of conflict did not automatically correspond to obtaining community acceptance. Conflicts, however dormant, have different expressions due to the socio-economic and cultural context [23,30,40,142] and companies often fail to account for the nuances of dissent and resistance in cross-cultural and socio-economical settings [30,43,143].

Based on our research, previously published interpretations of the SLO concept may not directly apply to the study area. A miscellany of causes such as the power imbalance between the local population and the company, the lack of developmental alternatives and community dependency, undermined what is the essence of a genuine social license to operate: the freedom to choose whether or not to accept a company or its project. When there is neither freedom nor alternatives, the indicators of an SLO may fail to represent the true nature of the relationship. These findings may apply to other remote Ecuadorian Amazon areas where socio-political and economic conditions are similar. Other studies have demonstrated a host of similar factors limiting community expression and choice concerning developmental plans, including: the absence of State oversight for social services and environmental monitoring [138,139]; disregard for prior consultation [65,66]; repression of dissent, which exhibits a clear power imbalance among the actors [70,109]; the communities' dependence on companies for social programs; and the general socio-economic condition of the rural populations [109,110,140,144].

A community's right to self-determination is undermined without the freedom to veto a company's operation [12,63]. This right may require regulating the decision-making process of projects and giving communities an active role in choosing their developmental paths [12,63]. This is in line with our findings that recognized the central role of procedural fairness in the process of SLO. To improve project procedural fairness and community self-determination, the local populations' decision authority appears important. Such authority can be supported by meaningful dialogue among parties, transparent communication about project impacts and benefits [4,30,67], and appropriate mechanisms to address community concerns, even when in the form of protests [11,67]. Community access to environmental information and adequate environmental controls should be granted by both the company and the State, to empower community input on projects [30,38,59]. Self-determination, as well as procedural fairness, is also intrinsically linked to FPIC rights [63]. These rights were legally established by the Ecuadorian constitution as a State's responsibility, but violations were reported by participants in this study, and mirror other studies based in Ecuador [65,66]. Therefore, a more effective FPIC procedure may be possible through synergistic collaboration between the company and the State [145]. Procedural fairness and self-determination may also occur when community dependence on companies is reduced; a condition which may be achieved if the State takes full responsibility for the management of social services in remote communities of the Amazon [30,109,146]. Instead, the Ecuadorian State continues to consider the Amazon as a territory that can be "sacrificed" in the name of national development; a territory whose control is entrusted to the oil companies and whose development possibilities seem restricted to oil extraction [109,110,140,147–149]. Remote communities desperately need viable developmental alternatives. Introducing such alternatives will require a complex and profound restructuring of the national economy [141] in order to imagine new trajectories of economic development toward more socially and environmentally sustainable energy sources [150,151].

On the whole, our results extended Moffat and Zhang's findings [4] on the role of procedural fairness, which appeared to be more central in the relational dynamics of SLO. Furthermore, self-determination was

highlighted as a key element in the context of structural power imbalances [23,63], common to indigenous communities in remote locations. Indeed, based on our findings, promoting procedural fairness and community self-determination, seem to be prerequisites for communities to express their views and decisions, and the minimum requirements for an authentic SLO.

*4.2. Limitations and Future Research*

This study presents some limitations. First, our limited sample size precluded cross-validating the structural model [130]. Future research may test the model on other samples to verify or modify as needed. Secondly, the model has left unexplored important aspects of participants' lived experience. Particularly, atmospheric quality and flora and fauna quality were not significant elements in the SLO process modelled. However, the qualitative data demonstrated that the perceived impacts of these environmental elements were a major concern for the study's participants. Future research may suggest modifications to the model to better fit participants' perceptions. Additionally, while investigating the SLO concept within an Amazonian indigenous context is a strength of this research, we acknowledge that there may be limitations. We involved a group of indigenous students as part of the research group and the questionnaire was translated in Kichwa. We acknowledge that any translation, both of concepts or linguistic elements, has limits in the adaptation of constructs to other cultural contexts [115]. Future research may consider exploring the SLO across indigenous and non-indigenous communities within the Ecuadorian Amazon, to compare findings. Finally, future research may consider exploring the SLO in other countries of the Latin America region, to verify the presence of similar conditions that limit the conceptualization and the development of a genuine SLO.

## 5. Conclusions

Social license to operate (SLO) is a concept that has been developed and adopted in mining intensive developed countries and is now gaining attention worldwide, including within Latin America. The present study was the first to investigate the SLO in an oil area of the Ecuadorian Amazon. Through household questionnaires and semi-structured interviews, a modified version of Moffat and Zhang's SLO model [4] explored relations among an oil company and a population of remote indigenous villagers. Based on this study, we agreed with those scholars that defined SLO as difficult to measure and model [9,32] due to the complexity of company–community relationships. Notwithstanding, the mixed methods approach we adopted supported a deeper understanding of the SLO process in the study area by giving voice to the participants and ensuring the results were grounded in their experiences. Compared to Moffat and Zhang's study [4], our model showed a more important role for procedural fairness in building trust and acceptance and approval. Indeed, procedural fairness was found to have a central role in mediating the effects on trust as determined by company provision of essential services (health and educational programs and housing availability), and the perceived quality of sustenance sources (i.e., fishing, hunting, harvesting, cultivating, and waterway quality). Moreover, results suggested that an SLO may not exist when a community is not free to exercise decisional authority over the company's operations; precluding the people's right to self-determination [12,63]. Findings indicated that to enhance the procedural fairness and community self-determination, it was important that the company engaged the local population in the decision-making process prior to and during the implementation of projects; promoted the villagers' access to environmental information together with the State institutions; fairly managed and distributed benefits among the villages; and addressed village concerns. In addition, other relevant aspects which appear to support procedural fairness and self-determination include the reduction of community dependence on the company, through the active engagement of the State in the provision of community social services and the development of economic alternatives to oil activities; and the fulfillment of FPIC rights through the synergistic involvement of both State and industry. To conclude, by supporting procedural fairness and community self-determination, communities are more likely to freely express their decision and provide a truer representation of a social license to operate.

**Author Contributions:** Conceptualization, A.D., M.D.M., T.E.P.; methodology, A.D., M.D.M., S.E.P., D.C.; formal analysis; A.D., T.E.P.; M.D.M., S.E.P., D.C., M.H.-R.; investigation, A.D., D.C., S.E.P., M.H.-R.; data curation, A.D., M.D.M., S.E.P., D.C., M.H.-R.; writing—original draft preparation, A.D., T.E.P., M.D.M., S.E.P., D.C., F.F., E.C.; writing—review and editing, A.D., M.D.M., T.E.P., S.E.P., D.C., G.D.F., F.F., E.C.; supervision, M.D.M. All authors have read and agreed to the published version of the manuscript.

**Funding:** This research received no external funding.

**Acknowledgments:** We thank the Universidad Estatal Amazónica of Puyo and the GIScience D4G Lab, ICEA Department, of the University of Padua, respectively for the important logistic support and the technical equipment for the field activities. We also want to warmly thank the group of students that took part in the research during the fieldwork: *ashka pakrachu*!

**Conflicts of Interest:** The authors declare no conflict of interest.

## Appendix A. The Questionnaire Items of the Latent Measures

*Appendix A.1. Environmental Quality*

How do you think these environmental compounds are changing in the proximity of your village in relation to the presence of the oil company? (from 1 = *much worse* to 5 = *much better*)

- Air
- Night light intensity
- Noise intensity
- Rivers and stream
- Soil
- Fauna
-  Flora

*Appendix A.2. Social Aspects*

How do you think these aspects are changing for your village in relation to the presence of the oil company? (from 1 = *much worse* to 5 = *much better*)

- Medical assistance
- House availability
- Access to recreational services
- Access to transportation infrastructure
- Access to electricity
- Employment opportunities
- Education
- Water to drink
- Water to cook
- Water to wash
- Fishing
- Hunting
- Cultivating
- Harvesting forest products

*Appendix A.3. Procedural Fairness*

To what extent do you agree with the following statements? (from 1 = *strongly disagree* to 5 = *strongly agree*)

- Before the implementation of new oil operations, the village is always informed about risks and impacts
- The village is always free to give (or not give) its consent to the oil company's operations without being forced in any way

- The oil company listens to and respects the village's opinions (adapted from Moffat and Zhang [4], originally from Tyler [57])
- The oil company autonomously inform the village about the conditions of the territory that could be affected by its activities
- When the village asks the oil company for information about the state of the territory, the oil company always provide this information
- The oil company can change its activities or projects based on the village's opinions (adapted from Moffat and Zhang [4], originally from Tyler [57])
- All villages in the oil extraction area receive fair compensation for the oil exploitation
- All families in your village receive fair compensation for the oil exploitation

*Appendix A.4. Trust*

Please, answer the following questions (from 1 = *total distrust* to 5 = *absolute trust*)

- Do you trust the oil company to keep its promises?
- Do you trust that the oil company is doing everything possible to avoid impacts on the environment? (adapted from Moffat and Zhang's item about responsible action of the company [4], originally from Tam et al. [117])
- Do you trust that the oil company is doing everything possible to avoid social impacts in your village? (adapted from Moffat and Zhang's item about responsible action of the company [4], originally from Tam et al. [117])
- In general, how much do you trust the oil company? (adapted from Moffat and Zhang [4], originally from Tam et al. [117])

*Appendix A.5. Acceptance and Approval*

Please, answer the following questions (from 1 = *not at all* to 5 = *very much*)

- To what extent do you accept the presence of the oil company in your territory? (adapted from Moffat and Zhang [4])
- To what extent are you in favour of the presence of the oil company in your territory? (adapted from Moffat and Zhang [4])

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
