# Peer review of "Is this a Real Choice? Critical Exploration of the Social License to Operate in the Oil Extraction Context of the Ecuadorian Amazon"

_sustainability, doi:10.3390/su12208416_

Round 1

Reviewer 1 Report

The paper has substantial potential to make a valuable contribution to the literature.  In particular, the paper demonstrates how key conceptualisations of Social Licence need to be adapted and updated to the context of the Ecuadorian Amazon. In doing so, the paper provides insights which may apply in other contexts where major power imbalances are present, distinct from the developed country contexts where SLO research had mostly been developed to date.

The paper goes into the intersections between procedural fairness, distributive fairness and informed consent. The paper claims that ‘a fair decision-making process leads to distributive fairness’. There is a link between these two factors however the level of direct causality claimed in this paper seems to go beyond my reading of papers cited here. For example, reference 10 states that these two are ‘often correlated’ which is not the same as saying one leads to the other.

Broadly, the methods seem sound. However, the paper provides insufficient explanation of the analysis procedure applied to the qualitative data (lines 360-361).  The process by which analysis occurred should be explained. This is particularly important because the paper draws heavily on the findings of the qualitative analysis.  Any steps taken to ensure the robustness of the qualitative analysis need to be outlined (e.g. whether the data were independently coded by different staff and the coding structures compared, or whether any software was employed to support the qualitative analysis).

Line 670. The points around pre-requisites are well made and extend the work of Zhang et al 2018 cited earlier in the paper as reference 51 which identifies ‘pre-conditions’. The manuscript could state here that it extends the work of Zhang et al 2018 by highlighting the importance of self-determination which has not featured heavily in previous SLO literature, and is crucial to acknowledge in contexts of substantial power imbalances.

The paper requires a comprehensive language edit.

At times the paper struggles with use of the definite article in English

In the abstract add ‘the’ to ‘…even in form of protest.’  i.e. even in the form of protest.

Page 4 Line 144 adjust wording ‘The FPIC was established…”  add something like  ‘use of’ or ‘relevance of’ i.e. “The use of FPIC was established…”

P 5 line 220 replace house with ‘housing’ in “…house availability and affordability”

P 10 line 454 replace ‘complicate’ with ‘complicated’

Line 705 The way ‘substance’ is used here doesn’t quite fit. Do the authors mean ‘subsistence’ ?

Line 634 Rephrase: “Based on our research, we assume the SLO concept may be difficultly applied in the study area”. From the context, I think the paper is trying to convey something like: “Based on our research, previously published interpretations of the SLO concept may not directly apply to the study area.”

Reviewer 2 Report

I enjoyed reading this paper and am happy to recommend that it be accepted with, or without, the minor revisions noted below.

Increasingly natural resource management issues are recognized as being characterized by not only complexity and uncertainty, but also multiple social viewpoints. Moreover, there is an acknowledgement that institutions and companies need not only regulatory permission but also “social permission” to conduct their business. The concept is recognised as an important consideration for sustainable development.  In a mining context, obtaining and maintaining a social licence requires the use of community-based approaches in which priority values are negotiated, rather than mandated.

Efforts to develop and maintain Social Licence to Operate (SLO), however, are often hampered by a lack of clarity around the different components that underpin these concepts. SLO is most often described in the literature as intangible and impermanent, subject to continual review and renewal by the different stakeholders involved. These extremely hard to define qualities also perpetuate the difficulty that institutions and companies have in defining the concept and their efforts to develop and maintain it. The paper seeks to address this issue directly and looks to provide more clarity and a deeper understanding of the SLO process and so represents a good topic for publication in Sustainability. The authors use a case study setting of an oil production area of the Ecuadorian Amazon, which provides a novel context in its own right – albeit with findings that can support similar SLO endeavours in other areas.

My own background and interests are as a participatory action researcher who works in the natural resource management area, and my comments are written from this perspective.

Title, abstract and keywords

These provide an accurate and useful outline, introduction and summary to the paper.

Introduction

This section introduces SLO well and sets the scene by using literature to expand on Moffet and Zhang’s model. The literature also provides additional context for the South American oil activity sector.

Materials and methods

The paper outlines the use of semi-structured interviews and a household survey.   The research method is sound and appropriate for the exploratory nature of the social research described. The authors provide a good explanation of how they developed their questions.

Results

Both qualitative and quantitative results are presented. These are clear and informative. I do not comment on the application of the SEM methodological analysis, but note that the results are in keeping with the qualitative discussion. The revised SLO model seems useful, and is well explained.

Discussion

The discussion section highlights the originality of the authors’ contribution to SLO. It highlights their conceptual model – raising the importance of procedural fairness - and its implications for indiustries and companies seeking to manage relations and engagement around SLO. It highlights the complexities involved in balancing different aspects of SLO, and how these interlinkages may play out in practice. And it reinforces the difficulties in utilising the concept to fully represent the complex nature of industry – community relationships. The paper also raise the importance of looking more closely at how the roles of the State, industry and communities interlink to influence SLO.

Conclusion

This section wrapped up the paper very well. In particular the authors highlight the importance of procedural fairness and community self-determination in developing SLO in practice.

References

The authors cite a useful and wide range of literature looking across SLO and other paper related fields. Moreover, a high number of these are recent indicating that the authors have brought in their insights from a range of literature – including current thinking. As an indication of this, 65 (more than 40%) of the 150 citations reference material written since 2015.

Editing

The paper read well. I did not read for editing but noticed a couple of errors – where I include sentences, it is suggested that these be rewritten to clarify the intent:

Line 310 – was (were)

Line 375 – eigen value

Line 402 – in regard to

Line 406 – During the study, we asked the ministry some details about two spill cases but we didn’t receive any document, despite the company suggested the ministry hold the information.

Line 408 –  Perhaps  indicative  of  this  control,  the  company  closely  monitored  our  project.

Line 411 – Many  [say who] complained  about  a  lack  of  consultation  prior  to  recent  company  developments,

especially those in closest proximity to the completed project.

Line 454 – complicate(d)

Line 470 –  [A] General  feeling  of  dissatisfaction  was  expressed  by  many  participants  towards  their  lack  of involvement  in the  decision-making process

Line 543 – was (were)

Line 605 – Regardless [of] reported  low  levels  of trust and acceptance  and approval, residents  expressed continued interest in social programmes provided by the company.

Line 634 – difficultly

Round 2

Reviewer 1 Report

The revisions have addressed all the points that I raised in the original submission. The paper does a good job of building on previous work in a novel application and context. The methods, analysis and discussion are all in good shape.